# Emerging Diamond Quantum Sensing in Bio-Membranes

**DOI:** 10.3390/membranes12100957

**Published:** 2022-09-30

**Authors:** Yayin Tan, Xinhao Hu, Yong Hou, Zhiqin Chu

**Affiliations:** 1Department of Electrical and Electronic Engineering, The University of Hong Kong, Hong Kong 999077, China; 2Joint Appointment with School of Biomedical Sciences, The University of Hong Kong, Hong Kong 999077, China

**Keywords:** NV centers, bio-membrane, fluorescent biomarker, optically detected magnetic resonance (ODMR), nanoscale sensing, quantum sensing

## Abstract

Bio-membranes exhibit complex but unique mechanical properties as communicative regulators in various physiological and pathological processes. Exposed to a dynamic micro-environment, bio-membranes can be seen as an intricate and delicate system. The systematical modeling and detection of their local physical properties are often difficult to achieve, both quantitatively and precisely. The recent emerging diamonds hosting quantum defects (i.e., nitrogen-vacancy (NV) center) demonstrate intriguing optical and spin properties, together with their outstanding photostability and biocompatibility, rendering them ideal candidates for biological applications. Notably, the extraordinary spin-based sensing enable the measurements of localized nanoscale physical quantities such as magnetic fields, electrical fields, temperature, and strain. These nanoscale signals can be optically read out precisely by simple optical microscopy systems. Given these exclusive properties, NV-center-based quantum sensors can be widely applied in exploring bio-membrane-related features and the communicative chemical reaction processes. This review mainly focuses on NV-based quantum sensing in bio-membrane fields. The attempts of applying NV-based quantum sensors in bio-membranes to investigate diverse physical and chemical events such as membrane elasticity, phase change, nanoscale bio-physical signals, and free radical formation are fully overviewed. We also discuss the challenges and future directions of this novel technology to be utilized in bio-membranes.

## 1. Introduction

Bio-membranes are ubiquitous and essential in sustaining life. All cells and tiny organelles are surrounded by bio-membranes. They not only act as the physical barriers that avoid potential microbial infection, but also regulate communications and transport inside and outside the cell [1]. Increasing evidence has demonstrated that the elasticity, electromagnetic, mechanical, and thermostatic properties of bio-membranes reveal diverse biological significances [2,3,4]. They control membrane behaviors, membrane signaling, functional membrane domains, and metabolism [2,3,4]. For instance, bio-membranes assist in transporting oxygen and carbon dioxide between the lungs and bloodstream to support our metabolism [5]. Those brain membranes affect our emotions because they possess receptors for signaling molecules such as dopamine [6]. As for membrane electrical signals, they express rich intra- and inter-cellular communications and regulate many biological processes such as the cell cycle and proliferation [7]. The flexoelectricity shows important implications for ion transport and hearing function [8]. The mechanical properties are closely related to the fundamental processes of cell differentiation, mitosis, and apoptosis [9].

It is thus of great importance to explore the bio-membrane behaviors, properties, and functions in biofields. The detection and introduction of nanoscale physical signals give us better insight into understanding and controlling bio-related behaviors. Specifically, local monitoring of the faint neural electric fields [10] enables not only the study of brain currents during cognition to improve neurodiagnostic systems, but also the identification of early stages of neurodegenerative diseases, such as Parkinson’s disease, Alzheimer’s disease, and other forms of dementia [9,10]. Applying a suitable electric field to cell membranes can promote cell proliferation and protein synthesis, as it controls the opening of signaling channels and promotes molecule exchanging from the extracellular matrix to the cytoplasm [11]. Static magnetic field stimulation may cause a significant increase in chondrocyte metabolic activity upon induction [12], making magnetic stimulation a potential treatment for cartilage recovery. In addition, the local energy transfer and chemical reactions can be reflected by the presence of localized temperature gradients and heat dissipation across bio-membranes [13]. Up to now, a number of measurement techniques, including optical and magnetic tweezers, traction force microscopy, and atomic force microscopy, have been extensively developed and applied to probe membrane behaviors and functions [14,15]. However, it is still quite challenging to develop suitable sensing probes with multitasking capabilities [15,16].

Among the various sensing protocols that have emerged over the years, one of the promising sensors in the bio-membrane field is NV-based quantum sensing [17]. It allows optical initialization and spin readout by the optically detected magnetic resonance (ODMR) technique [18]. Moreover, the spin energy levels are sensitive not only to electromagnetic fields but also to temperature and strain variations [19,20,21]. These specific properties, together with the photostability at room temperature and the excellent biocompatibility [22], promote NV centers as a prospective candidate for biological applications. In this review, we will focus on the emerging NV-based quantum sensing techniques applied to study the behaviors, signaling, and properties of bio-membranes. Section 1 will introduce a tailored view of bio-membrane structures, their biophysical properties together with functions, and the current detection methods for bio-membranes. Section 2 focuses on the introduction of NV-based quantum sensing protocols and the related major influential factors, as well as possible application scenes in bio-field. The latest developments of NV-based quantum sensing in bio-membrane systems are summarized in Section 3. The final part gives the overall comments on the current limitations and potential prospects of NV-based quantum sensing techniques.

## 2. Overview of Bio-Membranes

As a typical bio-membrane system, the cell membrane is the physical boundary and key interface between the cytoplasm and the external environment. It provides a stable intracellular environment for metabolism and a selective path for transmitting substances and signals between the extracellular environment and intracellular matrix [23,24,25]. In this section, we will introduce the typical structure of plasma membrane and its fundamental functions (Figure 1). Then, we focus on its structure-related physical properties, including the mechanics, electromagnetic, and thermal characteristics. The commonly used bio-membrane-related sensing methods are also mentioned and summarized.

### 2.1. Structures and Compositions

As is well-known, the fluid-mosaic model is widely accepted to describe the plasma membrane structure, where the essential components such as cholesterol and glycoproteins form the different specialized membrane domains, lipid rafts, and protein/glycoprotein complexes [26,27,28]. Most of the specific membrane functions are performed by membrane proteins, showing the characteristic functional properties of each type of membrane in the cell [29,30]. One typical example is the ion channel, a highly functionalized transport complex embedded in the plasma membrane. It can generate membrane potential as electrical signals to provide an important pathway for signal transmission in organisms. The function loss and disruption of ion channel can lead to various diseases, for example, cardiac arrhythmias and Long QT Syndrome [31,32,33]. Therefore, monitoring ion channels’ function has practical significance for diagnosing these diseases.

The fluidity, derived from the specific lipid bi-layer structure [34], allows the integral proteins to alter their conformational equilibrium, thereby improving their affinity toward ligands and achieving energy-efficient conformation dynamics [35,36]. Membrane fluidity is closely related to the processes of cellular force generation and balance, affecting physiological behaviors, such as cell adhesion, proliferation, and motility [37,38].

### 2.2. Biophysical Properties of Bio-Membranes

#### 2.2.1. Mechanical Properties of Bio-Membranes and Available Detection Methods

Mechanical characteristics are the paramount biophysical properties of the plasma membrane for regulating the mechanical integrity of the whole cell body [39]. As the primary shield of the cell, the plasma membrane determines how the external force is perceived and transmitted into the cell [40]. In living organisms, the membrane’s mechanical properties highly depend on the membrane–cortex–skeleton-based three-layer structure [41]. Membrane elasticity is the leading indicator determining how cells resist shape deformations and changes [42]. Overall, the plasma membrane displays low shear modulus, low viscosity, variable bending moduli, and high elastic modulus [43,44]. Another core mechanical characteristic of the plasma membrane is membrane tension. As a mechanical regulator, membrane tension governs many cellular events such as cell adhesion, migration, and membrane trafficking [45]. Increasing membrane tension inhibits endocytosis but leads to exocytosis [46]. When membrane tension exceeds a certain threshold, it can retard the polymerization rate of actin and depolymerize the actin filaments, leading to a decreased adhesion intensity and membrane tension [47]. Alteration in the mechanical properties of the plasma membrane is highly relevant to the development and progression of human diseases such as cancer. Therefore, the mechanical characteristics of the plasma membrane could be employed as an auxiliary indicator for disease diagnosis and treatment [48,49].

Probing methods to quantify the mechanical properties of plasma membranes have been fruitful over the past few decades. Generally, they are classified into three broad categories: (1) to monitor the substrate deformation to estimate the force such as cellular traction force microscopy (TFM) [50] and micropillar-based force-measuring apparatus [51]; (2) single-molecule force spectroscopy using instruments such as an atomic force microscope (AFM) [52] or magnetic/optical tweezer systems [53,54]; (3) molecular tension-based fluorescence microscopy (MTFM) with the help of force-sensitive fluorophores [55]. The TFM or micropillar-based force sensor is used to measure the deformation of the substrate via the images of fiduciary markers or micropillars before and after the cell force loading. Then, the deformation information of the fiduciary markers will be converted into the cellular force [56]. These methods offer a simple and flexible way to measure the force effects during various cellular events. The AFM and magnetic/optical tweezers are high-precision instruments that allow us to probe the strength of nanoscale protein-protein interactions or the force response of individual molecules [57]. Therefore, these methods are mainly used to study the local nanoscale membrane mechanics. The development of MTFM was propelled by the discovery of fluorophore pairs that can be separated by force-responsive molecules. With the help of Förster resonance energy transfer (FRET), the fluorescence signals emitted by such a pair enable us to estimate the separation distance of the molecules and the corresponding force applied [58]. Although these techniques have been well established as the standard protocol for membrane mechanics studies, these technologies’ limitations should not be ignored and need to be improved. For example, optical tweezers may introduce cell damage due to thermal effects and the phototoxicity of lasers [59]. Traction force microscopy and magnetic tweezers cannot achieve high spatial resolution [60,61]. The applicable force range of atomic force microscopy is limited, and the results require complex fitting [62]. The MTFM suffers from photo-bleaching of fluorophores [58].

#### 2.2.2. Electronic Properties in the Bio-Membrane and Available Detection Methods

The resting potential and action potential are typical electronic properties of membranes. Relative membrane permeability to K^+^ is the essential factor that produces resting membrane potential, while the action potential is generated by Na^+^ rapidly entering cells under the transient activation of Na^+^ channels, followed by an increase in K^+^ transport out of cells [63,64]. The action potential is a regenerative electrical phenomenon in living organisms, and it regulates many physiological processes including cell proliferation and differentiation (for example, inducing mitosis) [65]. In the nervous system, it can transmit electrical signals over long distances without attenuation. As a result, it is another indispensable method of biological signaling [66].

It is imperative to monitor signals of ion channels and membrane potential accurately and efficiently for studying neural signaling and other electrical excitation events. Membrane potential recording traditionally relies on electrode-based probing protocols: voltage-clamp and patch-clamp technologies, standard protocols for studying the characteristics and state of ion channels [67,68]. Compared with voltage-clamps, patch-clamps can monitor the electrical activities of an individual ion channel. This enables precise readout of membrane potentials with a high temporal resolution and signal-to-noise ratio [69]. However, their drawbacks are also apparent: both techniques are mechanically invasive, disrupting membrane integrity and dysfunction in electrophysiology. Besides, they are not capable of multiplexing and high-throughput recording [70]. Alternatively, molecular-level optical voltage imaging and calcium imaging can monitor electrophysiological activities in a more friendly way. Both techniques detect membrane potentials by monitoring changes in the fluorescent signals [71]. Common fluorescent tools include organic chromophores, fluorescent nanoparticles, quantum dots, endogenous fluorophores, genetically encoded proteins, and hybrid voltage indicators [72,73]. The calcium imaging method reports the calcium-associated spiking activities with calcium indicators, which could coordinate with Ca^2+^ and help quantify the free calcium concentrations [74]. Due to the excellent sensitivity of calcium indicators, calcium imaging is a valuable tool to monitor the electrical activities of both neuronal networks and individual neurons. However, because of their saturation effects and significantly delayed calcium flow dynamics (compared to the changes in action potential), calcium imaging is highly restricted in recording transient action potential and the high-rate firing of neurons [75].

Different from calcium imaging, the optical voltage imaging method depends on the fluorophores that are electrically sensitive. Voltage imaging protocols can capture electrophysiological events with sub-millisecond precision and micron-scale resolution. They can read out the electrical activity of each neuron in neural signaling circuits [76]. However, most fluorophores are easy to bleach in complex cellular environments, making them fail in long-term stable imaging. Additionally, organic fluorophores usually introduce physiological toxicity or pharmacological effects, and cell photodamage induced by prolonged high-intensity laser irradiation cannot be ignored. Moreover, most chromophores can be nonspecifically absorbed to the plasma membrane, introducing extra background noise. Genetically encoded indicators could specifically target the membrane receptors, but the complex transfection procedures always restrict their wide applications [30].

#### 2.2.3. Thermal Properties of the Plasma Membrane and Available Detection Methods

Temperature plays a crucial role in modulating cell function and behavior at a molecular level. Specifically, temperature determines the arrangement of lipid molecules and affects the thermal stability of the plasma membrane. When the temperature is higher than that of the source organism, the integrity of the plasma membrane might be destroyed due to lipid bilayer rearrangement [77]. For instance, hemolysis occurs when red blood cells are exposed to an environment above 37 °C [78]. Lipid membranes undergo phase transitions and phase separation when the temperature fluctuates. In a typical phase transition process, phospholipid bilayers go through a lamellar crystalline (subgel) phase (L_C_)–lamellar gel phase (L_β_)–lamellar liquid crystal phase (L_α_) transition process upon heating [79]. Beyond that, the mechanical characteristics of the plasma membrane are also highly related to temperature. Changes in temperature can generate thermal fluctuations of the plasma membrane, triggering large-scale deformation of cells [80]. The plasma membrane exhibits more elastic properties as the temperature decreases [81]. By affecting the transmembrane movement of ions, temperature can dominate action potential generation, thus affecting neural signaling processes [82]. Therefore, precise temperature detection is essential for monitoring the normal physiological function of cell membranes.

At present, cellular temperature measurement techniques at the macro/nanoscale can be divided into two categories: contact thermometer and non-contact thermometer [83]. A non-contact thermometer mainly refers to temperature-sensitive fluorescent sensors. They work through the principle that the sensors’ fluorescent properties (intensity, fluorescence lifetime, maximum emission wavelength, etc.) are quantitatively associated with temperature fluctuations [84]. Similar to electro-sensitive luminophores, temperature-sensitive luminophores contain organic and inorganic luminophores, including organic small molecule dyes, luminescent bio-macromolecular proteins, luminophore-embedded polymers, fluorescent quantum dots, luminescent metal clusters, etc. [85]. Luminescence-based thermometers can realize relatively high temporal–spatial temperature resolution and high-throughput data acquisition. However, in practical applications, fluorescence-based temperature sensors significantly depend on the concentration and distribution of fluoroprobes and suffer from photobleaching effects [86]. In addition to non-contact methods, the thermocouple thermometer, a representative contact thermometer, also provides an alternative solution to measure the local temperature of the plasma membrane. Based on the Seebeck effect of thermoelectric materials, a thermocouple thermometer reads out the local temperature of cells by converting thermal signals into electrical signals. Usually, this requires precise micromanipulation to target membranes with probes before testing. In this way, people can achieve a high-temperature resolution monitoring in living cells, but the spatial resolution is sacrificed [85].

## 3. The Rising Diamond Quantum Sensing

As previously described, the properties (mechanics, thermodynamics, and electromagnetics) of biological membranes are crucial for them to undertake a series of cellular activities and protective responses. Nevertheless, those properties are usually difficult to model and measure quantitatively, especially in the complicated cellular environment [87,88]. The demand for a general and nanoscale fluorescent probe that is stable and biocompatible for biological utilities has given rise to a variety of proposals for applications [89]. As emerging quantum sensing technologies, crystal defects in diamonds are promising candidates and have recently been attracting considerable attention in a variety of biomedical applications, owing to their solid stability and interesting optical properties [90]. Due to the intrinsic properties of diamonds, they also exhibit a high refractive index, robustness, and inertness, with low electrical but high thermal conductivity [91]. The certain crystal defects such as the prominent NV centers in diamond lattices show intriguing abilities, such as the unique optical addressability and spin properties that can be accessed by optical means at room temperature. Based on the quantum mechanical interactions of defects’ spin states, they can be extensively utilized in localized detection and measurement of nanoscale physical quantities such as magnetic fields, electric fields, temperatures, and strains [92]. Here, the emerging NV center-based quantum sensing technology is presented. We give a concise basic overview of the NV-center embedded diamond properties and how NV centers are suitable for nanoscale quantum sensing, specifically in bio-membrane fields, including the physical quantities that may be measured and the commonly used protocols for measurement.

### 3.1. NV-Center Based Quantum Sensing

The diamond material hosting NV centers shows superior properties as aforementioned and is an ideal platform for quantum sensing technologies, as the hosted NV centers can preserve their quantum features of spin-dependent photoluminescence even at room temperature [91,93,94]. The commonly used diamond materials are fabricated in different forms, including nanoparticles [95], bulk thin films or plates [96], and customized photonic structures [97,98], shown in Figure 2a. The NV center (a nitrogen atom substituting a carbon atom and an adjacent lattice vacancy) in the diamond crystal acts as a solid-state spin qubit and shows quantum behavior at room temperature (Figure 2b) [99]. The NV axis is defined as the vector from the vacancy towards the nitrogen atom, applicable to make the system theory understandable. The NV center in nanodiamond possesses high biocompatibility and photostability [17] and can be positioned several nanometers close to the cell membrane. These properties thus allow a subcellular spatial resolution for biological applications. In particular, the electronic spin states are magnetically, electrically, and thermally sensitive to the external perturbations with long coherence lifetimes [20,100]. Furthermore, NV electronic spins can be probed with microwaves and then optically read out to provide spatially resolved maps of nanoscale magnetic fields and other physical quantities measurement in a simple way [92]. The fluorescence of NV centers can be simultaneously quantified on the diamond surface, easily viewed by confocal or wide-field imaging microscopy, and recorded as the image of a two-dimensional magnetic field with an adjustable spatial pixel size [92,101]. The measurement protocols of NV centers are applicable under a wide range of conditions in broadband and narrowband field imaging, from DC to GHz frequencies and from cryogenic temperature to well above room temperature. These outstanding optical and spin properties, as well the simple and extensive measurements, along with the excellent biocompatibility, make the NV-based sensors promising candidates for a broad series of bio-oriented applications [18,89], specifically in bio-membrane fields. For instance, it has been reported that NV centers can be used to quantify the membrane elasticity [102] or the intracellular thermal gradient due to biological processes [103].

Serving as broadband detectors, NV centers can be modulated in an optical way due to their spin-dependent photoluminescence. It has three electronic levels, i.e., a ground state ^3^A, an excited state ^3^E, and an intermediate metastable singlet state ^1^A. The ground and excited states are spin triplets and are split into three spin sublevels: ms = 0, −1 or +1. The simplified energy diagram [18,99] is presented in Figure 2c. The NV center can be easily excited by a typically used green laser (~532 nm). During the excitation, it undergoes an optical transition from the ground state to the optically excited state ^3^E. The NV system then decays via two ways, the fast fluorescent emission from the ms = 0 of ^3^E and non-radiative transition from ms = ±1 of ^3^E to the long-lived singlet state ^1^A (Figure 2c). This makes the ms = 0 and ms = ±1 states distinguishable by fluorescent differences, where the ms = ±1 states are optically darker than the ms = 0 state. Through fluorescent emission, the NV center emits red photons within a broadband emission range (~650 to ~800 nm, see the absorption and emission spectra in Figure 2e). The emission wavelength falls right in the near-infrared window suitable for biological tissue detection by most biosensors, where the emitted light has its maximum depth of penetration in tissue [105]. The NV spin transition energies between these states can be determined by sweeping the microwave (MW) energies. When the applied MW is in resonance with the transitions from the m_S_ = 0 to ms = ±1 spin state, the fluorescence drops, shown as a dip in the fluorescence intensity (see Figure 2d) in the electron paramagnetic resonance (EPR) spectrum. Specifically, when there is no external magnetic perturbation, the dip tip falls at 2.87 GHz, the zero-field splitting parameter denoted as D. As a magnetic field is applied, two resonances appear in the EPR spectrum because the degeneracy between the ms = ±1 level is lifted (Figure 2c). The frequency separation between the two resonances is given by 2γB, where γ = 2π × 28 GHz/T is the electron gyromagnetic ratio and B is the magnetic field parallel to the NV axis (Figure 2d). Thus, measurements of the MW frequency immediately yield the absolute value of the magnetic field. This ODMR effect is characteristic of NV centers. To understand how different perturbations affect the spin energy levels and cause shifts in the EPR frequencies, the NV center’s spin Hamiltonian value can be inspected for reference [99,106]. As analyzed in the Hamiltonian function, magnetic and electric fields directly affect the spin with a vectorial dependence. Thus, the NV sensing can measure the magnetic and electric changes in both magnitude and orientation. Energy levels may also be perturbed by several other physical quantities via the zero-field splitting parameter *D*. Examples of quantities affecting *D* are pressure, strain, and temperature. Therefore, to sense the different physical quantities that affect NV centers is to measure the energy differences of those states and to detect transitions that occur between them (see below).

Based on the above, the NV centers may be influenced by different external properties such as magnetic fields, electric fields, temperature, or the presence of certain chemicals in the surrounding area, allowing for the versatile and multifunctional sensing of different physical quantities. Thus, one can read out these physical signals by NV centers with a microscope at room temperature, without the need of conventional magnetic resonance equipment.

### 3.2. Enabled Bio-Oriented Sensing Quantities

As described, the NV centers provide versatile sensing options that can be flexibly applied in bio-membrane fields. How the magnetic resonance signal of the NV center can be used to measure other useful quantities such as the electric field or temperature is described in the sections below. It can also be applied in orientation sensing. This multifunctional sensing strategy has high potential in solving many significant bio-related issues such as revealing the weak electromagnetic fields generated by neurons (or other excitable cells) or measuring the intracellular thermal gradient caused by biological or chemical processes. It may open a wider pathway for bio-membrane related applications.

#### 3.2.1. Magnetic Field Sensing

*Static Magnetic Fields.* Under the influence of a static magnetic field B, the Zeeman effect induced can affect the spin state of the NV center (Figure 3a), where the ms = ±1 sublevels are split [8]. The extent to which the resonance frequencies will be apart depends on the intensity of the external magnetic field. For external magnetic fields which are far smaller than 100 mT, the zero-field splitting is the dominant factor. In bio-membrane systems, the inner magnetic fields generated are far below 100 mT but higher than 0.2 mT [107]. In this scenario, the measurement protocol for the magnetic field can be simplified as indicated below, where the magnetic field projection along the axis of NV centers can be directly measured [104].
(1)$$v_{\pm }=D\pm \gamma \times B_{∥}$$

Here, *D*, γ, and $B_{∥}$ are the zero-field splitting, the electron gyromagnetic ratio of the NV center, and the magnetic field projection along the NV axis. In bio-systems, there are various natural static magnetic sources that are relevant to bio-sensing. For instance, natural biogenic minerals such as ferrimagnetic magnetite exist widely in organisms, which may strengthen tissues or indicate certain diseases [108,109,110]. Moreover, it is significant to detect localized magnetic fields to track artificially fabricated bio-probes such as magnetic nanoparticles, and this has been extensively applied in drug delivery [111], magnetic imaging [112], and magneto-genetics [113]. The static magnetic fields generated by those magnetites inside organisms or magnetic nanoparticles attached to bio-membranes can be measured or imaged by NV magnetometry at room temperature in principle.

*Fluctuating Magnetic Fields*. The more complicated ones to be measured in bio-systems are fluctuating magnetic fields, such as the transition metal ions [114], magnetic proteins [115], and free radicals [116]. They are abundant inside bio-membrane systems and are in dynamic change. Lacking proper detection methods, it is usually hard to detect those magnetic fields due to the relatively large frequency range of fluctuations and short-lived nature of molecules such as free radicals, resulting in limited understanding of those magnetic sources. The emerging detecting technology based on NV relaxometry (quantum relaxation microscopy) has subcellular resolution and can tackle the issues mentioned above [8,117,118]. Unstable magnetic species can be trapped at the diamond surface and detected by transducers such as spin traps [119]. The magnetic fluctuations can be measured by quantifying the effect of the life time (T_1_) of the NV magnetic states. Before disturbance by external perturbations, the three states (ms = 0, ±1) in the NV center are in equilibrium. During the T_1_ measurement, the NV center is firstly initialized by laser illumination to bring it in the ms = 0 state. Then, it takes certain amount of time T_1_ for the NV center to leave the ms = 0 state and return to the equilibrium situation, which is NV relaxation. By interacting with the surrounding perturbations, the NV electronic spin can relax faster, and this T_1_ relaxation time can be measured for the number of spins in its surrounding by detecting the brightness drop. The T_1_ scheme (Figure 3b) is frequently applied in bio-related sensing [120]. It can be determined by measuring the fluorescence of NV centers versus the waiting time τ (Figure 3b) between the two laser pulses [8,121]. In practical use, this method can be applied in imaging the stochastic magnetic fluctuations generated by gadolinium ions inside the HeLa cell membrane to achieve high-resolution bio-magnetic imaging [120]. This demonstrates that this real-time spin sensing method provides a minimally invasive tool for exploring dynamic magneto processes in bio-membrane systems.

#### 3.2.2. Electric Field Sensing

The Hamiltonian that describes the NV spin interaction with electric fields was derived from the molecular orbit theory [122], where the natural spin states are possible to be observed only in the presence of both the magnetic and electric field aligned with the NV axis. Typically, the frequency shift induced by the electric field is much smaller than that caused by a magnetic field. As such, to measure the electric-induced effect caused by the Stark shift reliably, it is essential to decouple the electric transitions from the fluctuating magnetic fields, providing an approach to switch the NV sensor from a ‘magnetic’ to an ‘electric’ sensing mode. The external electric field has been demonstrated to induce shifts in NV spin sub-levels, which have been measured at the atomic scale as reported (Figure 3c) [123].

Electrical fields in bio-systems have a time-varying existence as membrane potentials such as neuronal action potentials. To detect the dynamics of neuronal action potentials, it is crucial to understand the mechanisms of biological neural networks and elucidate the functions and biological features of the human brain. Nanodiamonds (NDs) can be embedded inside the cellular membrane (Figure 3c) where they can detect a strong potential drop between the extra- and intra cellular membrane space [8]. This allows the monitoring of neuron excitation in real time. NDs have been integrated into an artificial lipid bilayer for the detection of the adjacent magnetic fluctuation [124]. The real-time cellular-level detection for membrane potentials with high temporal and spatial resolution may reveal more significant findings related to cell activities.

**Figure 3 membranes-12-00957-f003:**
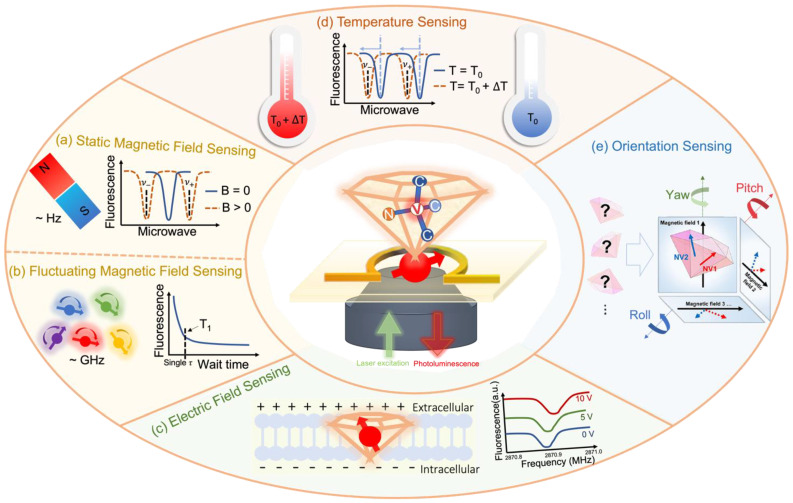
Bio-oriented nanoscale sensing for various physical quantities enabled by versatile NV-based quantum sensors. (**a**) Static magnetic field sensing using NV centers in diamond with a frequency of ~Hz, recorded by continuous-wave ODMR technique [17]. (**b**) Fluctuating magnetic field sensing using NV centers with a frequency of ~GHz, recorded by the longitudinal spin relaxation curve, where the pulsed sequence is used [17]. (**c**) Electric field sensing. Left is the schematic of the possible membrane potential measured by the NV center and right is the observed shift of ODMR resonance lines for different voltages [123]. (**d**) Nanoscale temperature sensing. Electron spin energy levels are mainly influenced by the temperature T, where a higher temperature makes resonances shift to lower frequencies. (**e**) Orientation tracking by NDs hosting NV centers. The schematic shows the determination of the orientation of an ND by ODMR, where the directions of multiple NV axes are projections to four different magnetic fields to rebuild the ND orientation in the diamond body space; it provides the 3D ND rotational tracking [125].

#### 3.2.3. Temperature Sensing

Another interesting aspect of the NV complex is its temperature-dependent spin properties. The zero-field splitting parameter *D*, caused by the spin–spin interaction in NV’s orbitals, depends on the diamond lattice length and is strongly correlated with the environmental temperature. As the local temperature increases, the diamond lattice spacing in the NV center is expanded, reducing the spin–spin coupling and lowering the zero-field splitting parameter *D* as well. Under ambient conditions, this influences the frequency *ν*_+_ and *ν*_−_ of magnetic resonances as shown in the following formula [126]
(2)$$\frac{dv_{\pm }}{dT}=\frac{dD}{dT}=-74.2\;kHzK^{-1}$$

This indicates that an increase in temperature makes both the resonance lines shift to lower frequencies (Figure 3d). The shift allows the measurement of temperature inside cells and bio-membranes. To realize the temperature sensing based on NV centers, the temperature-dependent zero-field splitting *D* requires that no other external field be present, which means that the external magnetic, electric fields, and strain should be small enough to be ignored. As reported by Neumann et al. [19], the influence from the axial magnetic and electric fields together with the strain can be diminished by applying a specific coherent control method [19]. Hence, the NV sensor can be potentially used as a high-precision nanoscale thermometer based on quantum-enhanced sensing, with a detectable and operational temperature range that extends from 100–1000 K [127,128].

As in bio-membrane fields, the local temperature gradient variations are closely related to a series of energy balances, biochemical reactions, and body temperature homeostasis inside and outside the cell [129]. For instance, a simulation model [130,131] presents a hypothetical variation in temperature inside the ion channels due to ions flowing across the cellular membrane at the initial stage of the action potential. More interestingly, temperature perturbations may greatly change the frequency of firing neurons, as has been demonstrated by Guatteo et al. [132]. Currently, NV-center-based quantum sensors are a more powerful method than conventional fluorescence probes in studying intracellular temperature variation thanks to their lasting photoluminescence together with high spatial and temporal resolution. The challenging local thermal variations can be measured by utilizing NV-based sensors. A temperature accuracy of 1.8 ± 0.3 mK with a resolution of 100 nm have been achieved [133]. The accurate measurements of localized temperature across bio-membranes may contribute to a deeper understanding of the patterns of transport and metabolic activities governed by cell membranes, and can help detect focal lesions as well as aid in targeted treatment clinically.

#### 3.2.4. Orientation Sensing

The long spin coherence time under ambient conditions and photo- and chemical stability make the NV centers a superior match for long-term tracking of bio-membranes. The NV-based orientation tracking is based on measuring the spin-dependent photoluminescence of the NV centers, similar to the static magnetic field sensing as mentioned. According to the orientation-dependent spin Hamiltonian, the rotational angles (roll−pitch−yaw) of the NV axis relative to the axis of an external magnetic field can be measured in the EPR spectrum (Figure 3e) [125]. For instance, the pitch angle (*θ*) follows the relationship
(3)$$\theta =cos^{-1}\left(\frac{h\Delta \omega }{\mu_{NV}B_{0}}\right)$$
where $\mu_{NV}$ is the magnetic moment of the NV center and Δω is the peak separation between the ODMR transitions [125,134,135].

In bio-membranes, the three-dimensional structure of molecular assemblies such as lipid bilayers are in dynamic motion to play their roles in various communicative processes or morphology maintenance [136]. The study of their characteristic movements, decomposed into the translational and rotational motions, is vital to elucidate the fundamental mechanisms of some basic bio-membrane-related processes, such as the cellular uptake of particles by the membrane or ion channel motions on the membrane [137,138]. The translational movements can be detected by various particle tracking methods, whereas the rotational (roll−pitch−yaw, Figure 3e) or orientational tracking of a single nanoparticle in living cells or membranes is very challenging using conventional approaches. Hence, the overall 3D rotational tracking of the single nanoparticle is hard to implement in bio-membrane systems, with considerable potential to explore the dynamic motions and rheological properties of bio-membrane systems. NV-based diamond sensing is a new approach applied as the single particle orientation tracking method. Recent reports show that NDs with NV centers act as promising candidates for precise tracking of 3D rotational motions [134]. The NV center as an orientation probe inside living cells was first demonstrated by measuring the rotation of NDs containing single NV centers [135]. Most recently, the 2D translation and 3D rotation of NDs have been tracked in biological systems using wide-field configuration [125].

## 4. Emerging Applications of Diamond Quantum Sensing in Membrane Systems

As mentioned in Section 2, NV-based quantum sensors can be applied widely and differently in bio-membrane fields regarding the related magnetic, electrical, thermodynamic, and mechanical quantities. Understanding the bio-function of the cell membrane requires the quantitative detection and analysis of the physical properties of the cell membrane. For instance, nanoscale magnetic field fluctuations derived from the fundamental spins are ubiquitous in biological systems and act as a rich source of information about the processes that generate them at the atomic and molecular levels, still waiting to be explored [124]. Nanoscale thermal gradient variations in the proximity of bio-membranes are also vital for understanding the cross-membrane motions due to biological or chemical interactions [139]. The following section will mainly demonstrate recent major bio-membrane-related applications from the physical and chemical perspectives to show the superior advantages of NV-based diamond quantum sensing applied in bio-membrane systems at atomic levels.

### 4.1. Measurements of Physical Events in Membrane Systems

#### 4.1.1. Membrane Structure-Related Measurements

Lipid Bilayers

Lipid bilayer membranes can serve as a useful and appropriate platform upon suitable modification for various bio-membrane applications such as single ion channel analysis or drug screening [140]. They can be utilized to develop probes for studying bio-membrane systems, with an emphasis on detecting the nanoscale molecules and atoms to gain information that may be hidden in the ensemble averaging. A certain type of nanoprobe is to be developed for sensing the weak magnetic fields that derive from essential spins in nanoscale biology, from naturally occurring (free radicals) or specifically introduced (spin labels). They are in great demand to study the dynamic processes in bio-membrane systems in situ, such as has been done by magnetic resonance techniques such as electron spin resonance (ESR) [141], where the sensitivity and resolution need to be improved. As shown in Figure 4a, Kaufmann et al. demonstrated the ND with a single NV as a nanoparticle probe, situated in an artificial lipid bilayer with gadolinium spin labels and acting as a nanoscopic detector for direct weak magnetic field sensing under ambient conditions with noncontact optical readout [124]. Specifically, the NDs with NV centers were surrounded by a supported lipid bilayer labeled with Gd^3+^ for producing characteristic magnetic fluctuations in the lipid environment (Figure 4(aiii)) as the detected target. Changes in the spin relaxation time (T_1_) of this single NV spin probe (Figure 4(av,vi)) located in the lipid bilayer were optically detected at a projected sensitivity of ~5 Gd spins per Hz^1/2^ [124]. This demonstrates that the NV quantum sensor is sensitive to cross-lipid magnetic fluctuations generated from a small number of spins such as Gd labels in the nanoscale vicinity of the ND. The detection of such a small number of spins in a biological lipid model brings new possibilities for bio-membrane information measurement. It also highlights the potential of NV diamond sensors as a magnetic probe in nanoscopic bio-membrane systems, circumventing the fundamental issues related to ensemble averaging.

Ion motions through ion-channels

Ion channels enable selective and passive diffusion of ions across the cellular membrane, while the ion pumps create and uphold the potential gradients actively across membranes in living cells [142,143]. Recent measurement techniques such as patch-clamp methods are invasive and hard to scale up [144]. The approach suitable for ion-channel monitoring is to consider a non-invasive and reliable detection method in situ. Based on the quantum properties of a single-atom probe, the NV-based quantum sensors are suitable and applicable.

As shown in Figure 4b, Hall et al. studied the quantum motions of an NV atomic probe tip near the ion channel and lipid bilayer [145]. The NV detector (Figure 4(bi)) was comprised of a diamond nanocrystal hosting an NV center fabricated at the end of an atomic force microscope (AFM) tip. It was locked and protected in the super stable diamond to be kept within nanometers of targeted samples for biological applications. To detect the weak fluctuations of magnetic moments arising from ion-channel operation, the quantum decoherence of the NV center induced by ion flux was measured, which shows an ultra-sensitive monitoring capability for ion-channel issues, far beyond the limits of magnetometer sensitivity [146]. In addition to the embedded ion channels, Hall et al. also set the lipid membrane and the instant surroundings as fluctuating electromagnetic sources and evaluated the effect of each source on the quantum coherence of NV centers quantitatively to identify the sensitivity of NV detectors to ion channel signals. Their theoretical findings show the possibility that ion channel operation can be detected in real-time with millisecond resolution by direct monitoring of the quantum decoherence of NV sensors. This may have great impacts on the fields of membrane biology and drug discovery.

**Figure 4 membranes-12-00957-f004:**
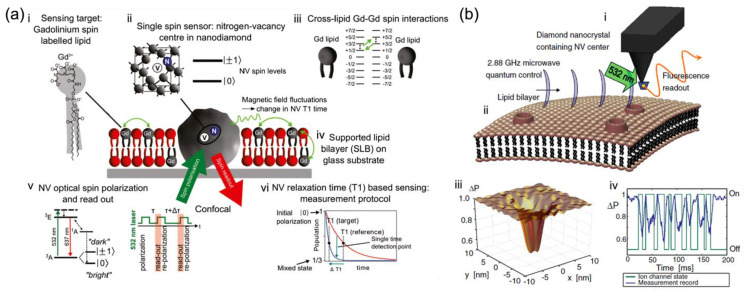
In situ nanoscale detection of dynamic processes in separate membrane structures by NV-based quantum sensing as nano-magnetometer probes. (**a**) Schematic of the single-spin NV sensor detecting nanoscale spin labels in a supported lipid bilayer (SLB) [124]. (**i**) Gd spin-labeled lipids are inserted into the SLB. (**ii**) Nanodiamond (ND) with a single NV optical center acts as a single-spin sensor. (**iii**) Magnetic field fluctuations generated by Gd spin labels influence the NV spin state. (**iv**) SLB is formed around an ND fixed on a glass substrate. (**v**) Electronic energy structure of the NV center showing optical spin readout of the spin sublevels m_S_ = 0 and m_S_ = ± 1and the T_1_ measurement protocol. (**vi**) Schematic of the T_1_ measurement. [124] (**b**) NV quantum imaging of the ion-channel operation [145]: (**i**) A single NV defect in a nano diamond is placed on an AFM tip. (**ii**) The cell membrane near the host channels permits the flow of ions across the membrane surface, resulting in an effective fluctuating magnetic field affecting the NV quantum spin state. (**iii**) The NV decoherence leads to a fluorescence decrease, most distinct in areas close to the ion-channel opening. (**iv**) Variations in fluorescence allow the temporal tracking of ion-channel dynamics [145].

#### 4.1.2. Membrane Motion Dynamics and Fluidity Measurement

Membrane motion tracking

The bulk lipid in the cell membrane makes a fluid matrix for protein rotation and lateral diffusion to achieve physiological functioning [147]. Such high membrane fluidity enables membrane proteins and molecules to diffuse rapidly in the plane of the bilayer and to interact with one another in cell signaling, which is vital for life [41]. Though it may seem hard to track the membrane fluidic motion due to the complex membrane model and the dynamic environment, Feng et al. still figure out a way by attaching diamond particles to live cell membranes for 6D tracking in measuring live bio-membrane dynamics [148]. Feng et al. demonstrated the synchronized 3D translation together with 3D rotation tracking of diamond particles based on NV magnetometry, and the simultaneous precision of ~10–40 nm for translation and ~1−5° for rotation with a 1-s measurement time was achieved. They carried out the 6D tracking of single diamond particles on a giant plasma membrane vesicle to characterize the translation and rotation on a lipid vesicle as a model system for cell membranes [148]. This provided clues to separate the intrinsic rotation of the diamond particle from the geometric effect due to parallel transport on a curved surface (the plasma membrane vesicle sphere). They further applied the 6D tracking technique to monitor single NDs on live cell membranes. The motion characteristics of the NDs on the cell membranes under various controlled conditions (normal, fixed, necrotized, or ATP-depleted) indicated that the rotation of the NDs is associated with cell metabolic activities. This technique expands the toolbox of single particle tracking and brings a distinct approach to issues where the analysis of translational and rotational correlations is critical.

Membrane elasticity measurement

The cell membrane, attached closely to its cytoskeleton, is crucial in regulating cell mechanics. The overall cell elasticity is an important parameter underlying the dynamics of diverse cellular activities [149]. The elastic property of the cell membrane can be measured through several contact methods such as optical tweezers and atomic force microscopy (AFM) [150]. Specifically, AFM indentation is the most widely utilized method to measure the cellular elasticity, but the local indentation data are usually complex and ambiguous to interpret [151], as they are related not only to the intrinsic cellular mechanical properties but also the contact features between the membrane and the tip, the uncertainty of which would lead to ambiguity in data interpretation.

Cui et al. reported a strategy of measuring the non-local deformation of fixed HeLa cells induced by AFM indentation (Figure 5a) with the assistance of the orientation sensing by NV centers in NDs [102]. The non-local deformation away from the indentation point is independent of the contact details but relies purely on the intrinsic mechanical properties of biological membranes, irrespective of local indentation uncertainties. This solves the above-mentioned issues in that it does not require the detailed local contact information between the membrane and the indentation tip, making it simpler for measurement. More spectacularly, the spin resonances of the NV center dependent on the magnetic field along the NV axis were applied to measure the orientation of the NDs on the membrane surface, thus achieving surface deformation mapping [102]. This represents a unique and simple way to provide sufficient resolution and precision for studying bio-membranes. Under controlled structural manipulations, this strategy was able to detect the competition between the elasticity and capillarity on cell membranes soaked in liquid and to evaluate the elastic modulus and surface tension simultaneously measured by NV-center-dependent orientation sensing [102]. It was concluded that if the capillarity was not considered, the membrane elastic moduli would have been overestimated as in most previous studies using local depth-loading data. In addition, after the depolymerization of the actin cytoskeleton structure, there was a reduction of both elastic moduli and surface tensions. These findings bring the first clear experimental observation of the elastocapillary effect by AFM indentation on cell membranes.

Lipid phase change with temperature

The cell membrane has different domains that are vital for cellular functions, such as molecule transport, communication, and metabolic interactions with the surrounding medium [2,152,153]. Those functional domains are compartmentalized by distinct phases of lipid membranes, resulting in extensive research on understanding their structural and dynamic characteristics. The phase behavior of a lipid bilayer shows the relative fluidity of the individual lipid molecules and how this fluidity varies with temperature [154]. Broadly, a lipid bilayer may exist as a liquid or gel phase across a wide range of temperatures. All lipids can undergo a phase change from a solid to a liquid at the characteristic temperature [155]. The lipid bilayer fluidity, characterized by the two-dimensional translational diffusion of lipid molecules [156], determines the fundamental property of lipids with different phases and hence the different domains. Recently, fluorescent probes were utilized for the detection of nanoscale diffusion and the identification of nanoscale domains [157]. However, they generally alter the mass and structure of the target molecule and deteriorate the dynamics to be observed [158]. To address these issues, NV center-based quantum measurement can be used as a label-free technique with nanoscale detection volume for direct diffusion measurement without causing perturbations in biological surroundings.

Ishiwata et al. investigated nanoscale phase change detection of lipid bilayers utilizing nuclear spin detection within a small interval (~10 nm) (Figure 5b), determined by the depth of the NV center [159]. It was observed that the NV center showed a variation of the relaxation time as a function of the change in temperature. By combining the Monte Carlo translational diffusion and molecular dynamics simulation, the translational diffusion constant varied from 1.5 ± 0.25 nm^2^/μs to 3.0 ± 0.5 nm^2^/μs when the temperature changed from 26.5 °C to 36.0 °C, depicting a phase change from the ordered to disordered [159]. The phase change in the lipid bilayer was observed directly for the first time by NV magnetometry, paving the way for the label-free identification of domains in the cell membrane to reveal the relationship between cell membrane dynamics and cell function [160].

#### 4.1.3. Membrane Potential and Polarity Measurement

Understanding of the biological neural network dynamics needs quantitative detection and analysis, which are vital for gaining insight into information processing in the brain. The information transferred in neurons is in the form of action potentials (APs), where the propagation of signals is crucial for intercellular communication. Magnetic fields arising from neuronal APs can pass through biological tissue largely without perturbations, making magnetic measurements of AP dynamics possible to be carried out outside the cell or even outside an intact organism [161]. However, it is challenging to achieve single-neuron spatial resolution and scalable measurements for functional networks or intact organisms.

Barry et al. demonstrated the non-invasive magnetic sensing of neuronal AP dynamics with single-neuron sensitivity by optically probed NV centers in diamond, which is applicable in intact organisms (Figure 5e) [161]. This approach enables precise measurement of AP waveforms from individual neurons and correlates magnetic fields with the AP conductive velocity, which determines the AP propagation direction directly by the fact that NV centers are sensitive to the AP magnetic field vector. More recently, Price et al. performed the NV sensing protocol both intracellularly by NDs (Figure 5c) and extracellularly by diamond plates (Figure 5d) with cultures of biological neuronal networks, which are highly interconnected for generating synchronized waves of endogenous neuronal activity [162]. The NV sensing protocols based on the ODMR technique and MW modulation of photoluminescence were performed under a widefield fluorescent microscope, enabling high-resolution spatial and temporal mapping of the endogenous neuronal activity under biological conditions.

**Figure 5 membranes-12-00957-f005:**
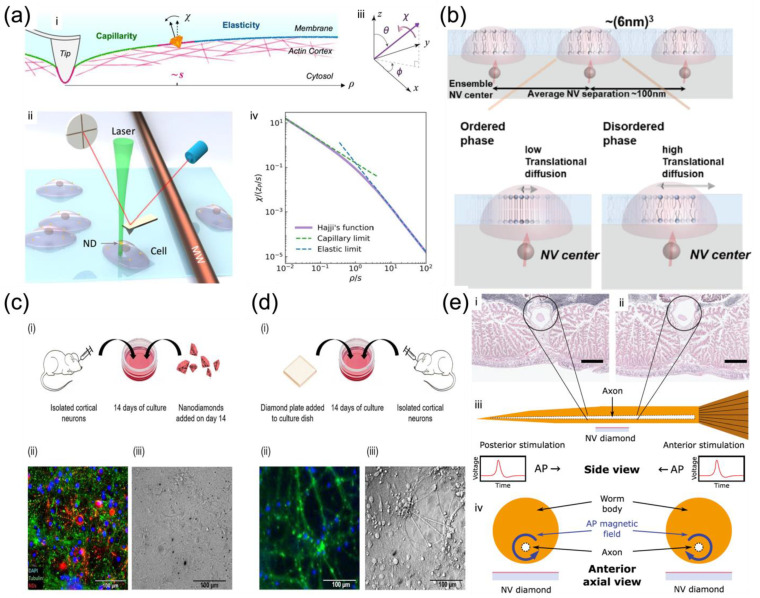
Biophysical properties of membrane systems measured by NV-based quantum sensing. (**a**) Simultaneous measurement of the capillarity and elasticity of the cell surface by ND rotation sensing [102]. (**i**,**ii**) Schematic of the AFM tip imposing a localized indentation on a fixed cell surface, with an ND attached nearby for rotation sensing measurement. (**iii**) The ND rotation is described by the direction of the rotation axis (*θ*, *ϕ*) and the rotation angle *χ*. (**iv**) The normalized rotation angle *χ* of the ND at a point loading as a function of *ρ*/*s*;. [102] (**b**) Schematic of the ensemble NV center detecting the phase change of the lipid bilayer [159]. (**c**,**d**) Schematic of the neuron culture with (**c**-**i**) NDs and (**d**-**i**) the diamond plate [162]. Fluorescent images show cells with NDs added on day 14 (**c**-**ii**) and cells after 14 days of culture on a diamond plate (**d**-**ii**). Differential interference contrast (DIC) of live cells with NDs ((**c**-**iii**) and live cells on the diamond plate (**d**-**iii**)) are presented [162]. (**e**) Magnetic sensing of the neuronal AP propagation exterior to the live organism [161]. Transverse sections of the midpoint of the worm illustrate the giant axon radius tapering from (**i**) smaller near the posterior to (**ii**) larger near the anterior. (**iii**) Schematic of the cross-sectional side view of the live worm and NV diamond sensor. (**iv**) Illustrative cross-sectional axial view looking from the anterior end [161].

#### 4.1.4. Nanoscale Thermometry in Membrane

Generally, the recent understanding of thermal effects in nanoscale biological systems is on the basis of macroscopic measurements [163]. Less is known about the nanoscale local thermostability or heat tolerance of subcellular organelles. Tsai et al. presented the hybrids of gold nanorod–fluorescent ND (GNR–FND) (Figure 6a–c) as a combined nano-heater and nanothermometer inside living cells [164]. With both heating and probing by the 594 nm laser, the temperature changes were measured by recording the spectral shifts of the zero-phonon lines of NV centers in fluorescent NDs [164]. This method is capable of identifying the rupture temperatures of membrane nanotubes in human embryonic kidney cells and generating high-temperature gradients on the cellular membrane for optically controlled hyperthermia. The results demonstrated a new paradigm for hyperthermia research and application.

Lanin et al. demonstrated fiber-coupled optical NV thermometry for individual thermo-genetically activated neurons based on ODMR technique [165]. The temperature variations from single neurons were generated by laser and were read out with the NV fiber sensor, strongly related to the fluorescence of Ca^2+^ ion sensors, serving as the real-time indicators of the inward Ca^2+^ current across the neuron cell membrane expressing transient receptor potential (TRP) cation channels (Figure 6d) [165]. This brings the possibility for measuring the neuronal activities with regards to the temperature in real time.

### 4.2. Measurement of Chemical Events in Membrane Systems

Chemical reactions attracting strong scientific interest are the cellular oxidation-reduction (redox) processes. Redox reactions are crucial in maintaining metabolic functions in living cells [166,167], such as energy generation, cellular respiration, and differentiation [166]. The local monitoring of redox reactions is critical for gaining insight into the cellular processes for maintaining cell viability, where the free radicals play a key role [168]. Detection of free radicals utilizing fluorescent NDs has recently been demonstrated. Barton et al. showed that the spin relaxation time of NV centers is sensitive to the number of nitroxide radicals, showing a resolution down to ~10 spins per ND (10^−23^ mol in a localized volume), with the NV centers in NDs coupled magnetically with nitroxide radicals inside a bioinert polymer coating [119]. This colloidally stable system can be used dynamically for spatial and temporal readout of the redox chemical process near the ND surface in the liquid environment.

As for cellular metabolic activities, Nie et al. also showed that the T_1_ spin relaxometry measurement of NV quantum defects can be applied in detecting free radicals and its generating process both in living cells and isolated mitochondria with subcellular resolution [169]. Fluorescent NDs were functionalized to target a single mitochondrium inside cells for metabolic activity monitoring.

## 5. Prospects: Limitations and Opportunities

NV-based diamond quantum sensing has spectacular properties in measuring the dynamic membrane physical quantities as well as the membrane-related processes at room temperature, with a simple optical readout via a conveniently modified microscope. The excellent photostability allows its usage as a stable bio-label. Practically, the localized measurement of nanoscale physical quantities such as magnetic fluctuations, electric signals, and temperatures inside or across the membrane has been demonstrated at the early stage. Besides the outstanding spin properties, researchers have also started to harness other optical properties to capture the rotational motions of single NV centers in NDs [170]. This orientation sensing capability of NV centers can be utilized to detect the membrane force and traction to a subcellular level, which undoubtedly will lead to a new path for a deeper understanding of membrane behaviors and motions.

Despite the superior properties and versatile applications brought about by NV quantum sensors, the problematic detection issues such as the fluctuating performance of spin properties in diamond materials, the restricted penetration depth of the excitation laser, and the relatively high MW power constrain their practical utilization in biological fields. As solutions, quantum sensing with fiber-coupled [171] or integrated devices [172], even in an MW-free manner [173], were recently developed in the bio-related field. Beyond that, to the best of our knowledge, there is no recent direct measurement of nanoscale mechanical signals in bio-systems by NV quantum sensing. The NV quantum sensors may open the door for measuring bio-oriented mechanical properties at a subcellular level and push the proof of concept to the next generation.

Other exciting developments are new pulsed protocols that allow selectivity for various samples or the capability of measuring chemical shifts [104,174], employing quantum lock-in detection to constantly probe the target signals. Efforts have also been paid to achieve atomic resolution [175]. Though the reported sensitivity of the diamond quantum sensors is sufficiently good in some cases, challenges still remain [176,177,178]. To be specific, imaging temperature variation mapping is challenging, as the temperature gradients may dissipate rapidly at microscales in most materials [92]. The diamond chip in contact may also accelerate the heat dissipation from the measured sample due to the outstanding thermal conductivity of diamond, hence affecting the accuracy of the temperature measurement. Another aspect is that the measurement of weak magnetic fields associated with human neuron activity still requires further improvements in the case of a single AP [179]. The concerns in the detection of neuronal fields and cardiac magnetic fields for diagnostic purposes may speed up the technological developments of quantum-enhanced biosensing based on NVs. Another challenge for diamond sensing is the electric field sensing. Generally, the electric interaction of NV centers is much smaller than the magnetic interaction. Therefore, the NV center is more sensitive to the magnetic fields rather than electric fields, the sensitivities of which require more improvements.

The success of these methods with further development will rely on high-quality diamond materials. Specifically, defect generation [180,181] and ND fabrication [182] are broadly studied to realize optimal coherence times. High-quality diamonds may open a broader world for biological applications. For instance, as aforementioned, the tiny cell tractions or forces can be detected based on the orientation sensing properties of NV complex in principle. Moreover, NDs could also be combined with other sensors to achieve multi-parameter sensing in the meantime. Specifically, the coupling of NDs with non-paramagnetic sensors (i.e., the calcium sensors) have high potential in deepening our understanding of the roles of radicals or the non-paramagnetic species in one biological process [183].

## Figures and Tables

**Figure 1 membranes-12-00957-f001:**
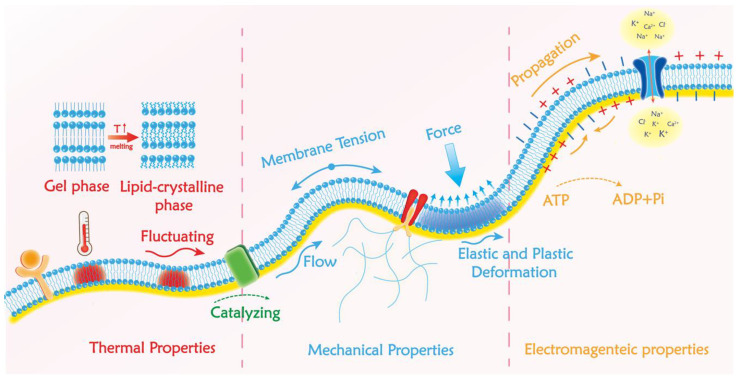
Typical structure of the cell membrane and its biophysical properties. Fluid mosaic model of the plasma membrane and the scheme of the three-layer composite of the membrane–cortex–cytoskeleton are presented; the basic thermal, mechanical, and electrical properties of the plasma membrane are marked.

**Figure 2 membranes-12-00957-f002:**
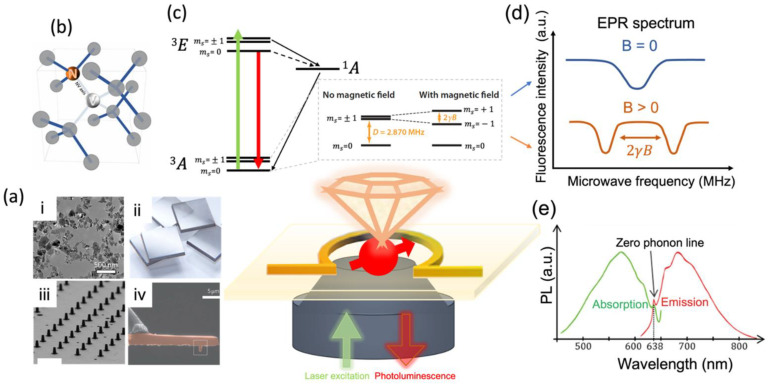
Optical illumination and characteristics of the NV center in diamond optically readout by a microscope for magnetic sensing and imaging. (**a**) Synthetic diamond materials with different modalities: (**i**) Transmission electron microscope (TEM) image showing the commercially available HPHT NDs [95]. (**ii**) Schematic image of millimeter-sized bulk diamonds that are commercially available [96]. (**iii**) Scanning electron microscope (SEM) image showing diamond nanopillars [97]. (**iv**) SEM image showing an all-diamond cantilever with a tip containing a single NV center [98]. (**b**) Schematic presenting the NV center structure: a point defect of a diamond composed of a nitrogen atom N substituting a carbon atom in the lattice and a vacancy (a missing carbon atom) right next to it [99]. The NV axis is defined as the vector from the vacancy towards the nitrogen atom. (**c**) Simplified energy-level diagram of the NV^−^ center: it has a ground state of symmetry ^3^A_2_, an excited state ^3^E, and a metastable singlet state ^1^A. The three spin sublevels are presented as m_S_ = 0 and m_S_ = ±1 with zero and nonzero magnetic field B. The red arrow indicates the radiative transition, resulting in the fluorescent emission. Black arrows indicate the strong and weak non-radiative decay via the metastable singlet state. *D* is the zero-field splitting parameter and 2γB is the Zeeman splitting, where γ is the electron gyromagnetic ratio [18,99]. (**d**) The electron paramagnetic resonance spectrum of the NV center at the zero and nonzero magnetic field, recorded by the ODMR technique. The degeneracy between the m_S_ = ±1 state is lifted, and the frequency separation between the two generations is determined by 2 γB. (**e**) Optical absorption and emission spectra of the NV center [104].

**Figure 6 membranes-12-00957-f006:**
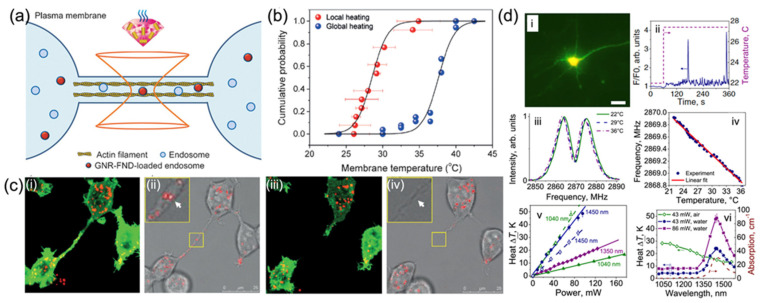
Membrane thermostability and temperature variations measured by NV nanoscale thermometry. (**a**) Schematic of the hybrid GNR-FNDs in the endosomes of membrane-tunneling nanotubes (TNTs) for both heating and temperature sensing activated by a 594 nm laser beam [164]. (**b**) Empirical cumulative distribution chart of the membrane temperatures where TNTs are ruptured by local heating and global heating of GFP-transduced human embryonic kidney (HEK) cells [164]. (**c**) (**i**) Fluorescence and (**ii**) merged bright-field with fluorescence images of HEK cells transduced with actin-GFP fusion proteins (green) and labeled with GNR–FNDs (red). (**iii**) Fluorescence and (**iv**) merged bright-field with fluorescence images of GNR–FND-labeled, GFP-transduced HEK cells after irradiation by a 594 nm laser with 330 mW for 6 s. Insets of (**ii**) and (**iv**): Enlarged views of the areas with the particle irradiated by the 594 nm laser enclosed in yellow. White arrows indicate the particles being irradiated [164]. (**d**) Fiber-optic thermometry for single thermo-genetically activated neurons [165]: (**i**) Fluorescence image of a laser-activated neuron. (**ii**) Fluorescence of the Ca^2+ ^sensor in a laser-activated neuron as a function of time (solid line) with the temperature within the laser-irradiated area inside the neuronal culture increased in a stepwise fashion (dashed line) for TRPA1-expressing neurons. (**iii**) Photoluminescence intensity from NV centers in a diamond microcrystal on the fiber tip, varies according to the MW frequency at 22 °C (green), 29 °C (blue), and 36 °C (maroon). (**iv**) The central frequency of the EPR spectrum of NV centers recorded by ODMR, changes with ambient temperature (filled circles) and its linear fit (solid line). (**v**) Laser-induced temperature change ΔT of water (filled circles, triangles, and diamonds) and the NV-diamond microcrystal in air (open circles and triangles) changes with different power of heating radiations with λ ≈ 1040 nm (triangles), 1350 nm (diamonds), and 1450 nm (circles). The linear fits are shown by solid and dashed lines. (**vi**) Absorption spectrum of water (dashed line) versus the temperature change ΔT induced by the optical radiation filled with water by 43-mW (circles) and 86-mW (rectangles), measured as a function of the wavelength λ [165].

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
