# Peer review of "Emerging Diamond Quantum Sensing in Bio-Membranes"

_membranes, 2022, doi:10.3390/membranes12100957_

Round 1

Reviewer 1 Report

The review from Tan et al. is an interesting piece, on a topic of interest for Membrane and about which a comprehensive literature review was needed.

I have some concerns with the introduction of the ODMR technique. I would suggest the authors to revise it to make it more rigorous and precisely worded. In particular, the authors call “EPR spectra” the ODMR spectra. A wording  choice that I found very confusing, and particularly problematic being present in a review that is intended to be used as a resource for other researchers that are not necessarily familiar with magnetic spectroscopies. See for example ref 112 and Gooverts 2017 (a review that I suggest to read and to cite when introducing the ODMR technique https://doi.org/10.1002/9780470034590.emrstm1524).

I think that panel 2c should be more clear. Only the spin sublevels of the T0 (3A) state are shown, whereas also the T1 and S1 states (3E and 1A, respectively)  should be shown and the transitions connecting them in order to show how a frequency scan affects the optical properties of the system. I think that otherwise a reader unfamiliar with double resonance techniques could hardly understand what is going on. Some of the reviews that they cite (ref 18 or 112) have good examples of such a figure.

Author Response

Thank you.

Reviewer 2 Report

The review from Tan et al. describes NV centers in diamond as a powerful tool for detection of several membrane properties using magnetic and optical methods.

The review has an extensive amount of references, and is clearly written. The system shows promise as the authors suggest and the ample amounts of references are convincing.

However, I have several issues that should be fixed to make it better for the reader.

1) Lenght.

The review is very long. Since several, highly generic concepts about membranes and some descriptions of the techniques are repeated several times. I would strongly advise the authors to trim down the repetitions to improve the overall readability of the review. Also, the authors occasionally write long sentences that exalt the virtues of the technique, but do not add anything to the scientific discussion, thse could be trimmed as well (as an example, see lines 18-19 in the abstract).

2) Theoretical description of the system.

The spin state is of paramount importance for the system. Just a minuscule diagram of the energy levels in the figure is not enough.

The spin system and energy levels should be very clearly defined from the start. The authors also make some confusion between spin states and spin sublevels which hurts the discussion and must be corrected (it is correct at line 451 and confusing at best at lines 321-322, for example). 

In addition, the concept of "the axis of the NV center" is repeated several times, but has never been clearly defined/discussed. 

3) Defining radicals and other paramagnetic species as "fluctuating molecular magnets" is just wrong. A molecular magnet is something different (either as single molecule magnets, like the manganese cluster, or the mixed organic-inorganic molecular based magnets). Also the discussion of paramagnetic species being the origin of fluctuating magnetic fields is somewhat misleading. The whole point would be more rigorous if rewritten in terms of fluctuating magnetic moments, perhaps.

Minor points:

Some sentences seem to be missing either the verb or the subject. Please re-read the manuscript carefully (for example lines 281-283) also for typos (line 188: and a followed increase of K, should be followed by an increase of K; line 387, orientated should be oriented).

Lines 293-297: the various properties of diamonds as thermal conductivity and robustness (already mentioned in the previous section) do not really seem to be connected to the "preservation of the quantum features of spin dependentphotoluminescence".

Line 302: "under" room temperature. Do the authors mean "below" or "at" room temperature?

Lines 457-458: either it is a cell membrane or an artificial lipid bilayer, they are different things... perhaps the authors are correct, but they should be clearer about the system.

Line 464-465. "ambient temperature" probably should be "temperature of the environment"

Author Response

Thank you.

Reviewer 3 Report

In this manuscript, the authors reviewed the recent progress of NVs 

in diamond as sensors for biomembranes. The NVs possess physical 

properties of spin and fluorenscence, which can respond to external 

influences, such as magnetic field, electric field, strain, and 

temperature. As a shield of cells, biomembranes separate inside and 

outside environments, and provide biofunctions, e.g. substance 

transport and signal transmission. Mechanical, electric, magnetic, 

thermal and chemical properties, as well as their detection methods 

are summarized. A proper outlook is also given. Besides, the 

references are also sufficient. As for the concept of quantum 

sensing, I did not see any aspect related to actual quantum physics 

or chemistry. It seems that all can be treated and understood with 

traditional physics or chemistry. In my opinion, it is not 

necessary to emphasize the quantum feature of NVs. In a word, this 

work is significant and suitable for the publication in Membranes.

Author Response

Thank you.

Round 2

Reviewer 2 Report

The authors have revised their review following the suggestions of all referees.

I still have seen a few minor English errors here and there, but they will likely be corrected in the editing/proof stage.

I would now say that it can be published without any further changes.